# High-Resolution Thermal Imaging and Analysis of TIG Weld Pool Phase Transitions

**DOI:** 10.3390/s20236952

**Published:** 2020-12-05

**Authors:** Nicholas Boone, Matthew Davies, Jon Raffe Willmott, Hector Marin-Reyes, Richard French

**Affiliations:** 1Sensor Systems Group, Department of Electronic and Electrical Engineering, University of Sheffield, Sheffield S10 2TN, UK; nick.boone@sheffield.ac.uk (N.B.); j.r.willmott@sheffield.ac.uk (J.R.W.); 2Department of Physics and Astronomy, The University of Sheffield, Sheffield S10 2TN, UK; h.marin-reyes@sheffield.ac.uk (H.M.-R.); rfrench@i3drobotics.com (R.F.)

**Keywords:** thermal imaging, welding, additive manufacturing

## Abstract

Tungsten inert gas (TIG) welding is a well-established joining process and offers the user flexibility to weld a large range of materials. Ultra-thin turbine tipping is an important application for TIG welding that is exceptionally challenging due to the wide range of variables needed to accurately control the process: slope times, arc control, travel speed, etc. We offer new insight into weld pool characteristics, utilizing both on- and off-line measurements of weld tracks. High-resolution thermal imaging yields spatially and temporally resolved weld pool phase transitions coupled with post-weld photographs, which gives a novel perspective into the thermal history of a weld. Our imaging system is filtered to measure a 10 nm window at 950 nm and comprises a commercial Sigma lens to produce a near-infrared (NIR) camera. The measured near-infrared radiance is calibrated for temperature over the range of from 800 to 1350 °C.

## 1. Introduction

Aerospace high bypass turbofans engines, such as the Rolls-Royce Trent 1000, have hundreds of individual compressor blades mounted on the eight-stage intermediate pressure engine section. The high temperatures and friction that these are exposed to during operation result in wear that must be repaired at a cost of up to GBP 7000 per blade.

Much work has been undertaken to create jet engines with good structural characteristics [1]. However, it is critical to improve the understanding of maintenance processes to complement the developments in design. During the operation of aerospace turbofan jet engines, the compression of atmospheric air and aviation fuel is performed by high-value metal alloy compressor blades. Extreme operational temperatures and friction result in wear and deformation. This necessitates entire engine overhauls every 30,000 h of operation. The turbine blades need to be removed from the engines for inspection and repair, where possible. The standard remanufacturing process for the turbine blades is conducted by layers of weld deposition or additive manufacturing (AM). The process of turbofan blade remanufacturing incorporates the gas tungsten arc welding technique (GTAW) or tungsten inert gas (TIG). TIG welding is a well-established joining process and offers the user flexibility to weld a large range of materials. Ultra-thin turbine blade tipping with TIG welding and an additive alloy filler are used to replace worn material, which is exceptionally challenging. This is because of the large number of variables needed to accurately control each part of the weld process. Generally, success is determined by the accuracy of the weld system, its slope times, the arc control circuit, linear arc travel speed, electrode angle and electrode material type.

The repair process is currently undertaken by highly skilled workers that weld layers of material onto the tip of the blade where the wear takes place. The part is then re-machined, heat-treated and coated to form the new blade edge. It is thought that 85% of blades recovered from service are repairable. However, only an estimated 45% are successfully repaired [2]. The high failure rate is attributed to human input in the process. Robotic welding coupled with novel sensing systems shows significant promise in closing the gap. The development of automated alternatives is paramount to reduce costs and improve reliability [3] and automation of the AM process. The robotic TIG system in this study was designed to give automated, repeatable and improved quality of welding for turbine blade repair. Machine vision systems work in conjunction with advanced sensory systems to generate data for the robotic computer numerical control (CNC) welding platform [4].

Problems can be caused in manufacturing processes by mechanical imperfections i.e., finite tolerances, contamination of the work piece, and equipment malfunctions. High frame rate, high-resolution thermal imaging now provide the missing data required for the robotic system to reach its potential as an Industry 4.0 additive manufacturing approach. This offers a map of the thermal field in two spatial dimensions unlike other online nondestructive test and evaluation techniques such as spectroscopic analysis. Spectroscopic analysis uses peaks in the emission spectrum of the weld to calculate electron temperature. Fluctuations in the electron temperature correlate with defect formation in the welding process. Visible imaging is also used to detect weld defects [5]. A high-resolution focal plane array imaging system sensitive to visible light is used to capture real-time images of the weld as it forms. These images can then be analyzed to detect defects [6]. The thermal imaging technique demonstrated in this work combines the benefits of temperature measurement with those of high-resolution imaging to provide data for a novel analysis technique for phase transition tracking and defect detection.

Thermal imaging is utilized in a diverse array of engineering research applications and industries. It has been ubiquitous with military applications since its conception, it shows immense promise as a medical diagnostic tool [7], and it is well established as a manufacturing process monitoring tool [8,9,10,11]. Thermal imaging cameras provide information about thermal gradients and relative temperatures within the field of view (FOV) of the imaging system. Quantitative temperature measurements can be made with reasonable uncertainty when systems are sufficiently calibrated and certain characteristics of the subject, such as emissivity, of the imaging are known.

The vast majority of commercially available thermal imaging cameras sold are sensitive to wavelengths of infrared radiation between 8 and 14 µm, long-wave infrared (LWIR). This spectral range affords excellent signal-to-noise ratio from sub-zero to hundreds of degrees Celsius. This is ideal for military and medical applications because the range includes body temperature and the operating temperature of military assets such as vehicle engines. The sight path between camera and object is relatively free from absorption bands due to water vapor and other atmospheric gasses in this wavelength band that would otherwise obscure a measurement. Thermal imaging is also used for smaller niche applications in other atmospheric gas windows, for example, between 3 to 5 µm (mid-wave infrared, MWIR) and 0.85 µm to 1.1 µm (near-infrared, NIR).

The demand placed upon thermal imaging often leads to a design trade-off between frame rate, resolution (number of pixels within the FOV) and temperature measurement (dynamic) range. Typical frame rates are between 7 and 60 Hz, typical resolution is 320 × 240 pixels, and a typical temperature range is −20 to 1000 °C. Frame rates can sometimes be increased by reducing the resolution so that fewer pixels are measured in a given time period. Resolution and frame rate can be increased by measuring at shorter wavelengths, where mature silicon based focal plane array (FPA) technologies are available. This improvement comes at the expense of increasing the signal-to-noise ratio at low temperatures and reducing the overall temperature measurement range.

NIR thermal imaging was developed during this research based upon the relatively new technology of scientific complementary metal-oxide semiconductor (sCMOS) cameras. These focal plane arrays (FPAs) can provide a high-resolution and simultaneous high frame rate for a wide range of temperatures. We demonstrated this technology by utilizing our system for thermal imaging in a novel robotic welding application. We believe the images of weld pool thermal dynamics are unprecedented and enable detailed analysis of the weld, substrate interaction and freezing points.

## 2. Materials and Methods

### 2.1. Welding Experimental Arrangement

The heat management system (HMS) provided a high-frequency, constricted, pulsed weld current. This was achieved by means of a VBC Instrument Engineering InterPulse IE175 that was calibrated to BS:EN 5050412008. The welding current output was maintained to an accuracy of better than +/− 2% of the set point, between 100% and 40% of the maximum settings. Inexpensive 316 L stainless steel plates were used a surrogate for the austenitic nickel–chromium-based super alloys that are used in turbine engines during our experiments. The stainless steel was cold formed into a turbine blade profile 1.5 mm thick and 60 by 40 mm in rectangular profile. The blade surrogates were mounted robotically and welded using system parameters determined in previous studies [9]. The camera was mounted perpendicular to the direction of travel (as shown in Figure 1) of the robotic arm that was used to move the blade surrogate beneath the stationary welding arc.

### 2.2. Camera Design and Imaging

The system was developed specifically for monitoring the robotic TIG welding process. The mounting position was determined, and the required camera FOV was calculated. The electromagnetic emission spectrum of the TIG arc was found to be concentrated in the visible portion of the spectrum. The waveband sensitivity of our instrument was defined by the need to capture the near-infrared portion of the electromagnetic (EM) spectrum while minimizing sensitivity to the visible EM radiation from the TIG arc [12].

The thermal imaging system consisted of a Hamamatsu (10 Tewin Rd, Welwyn Garden City AL7 1BW) ORCA Flash 4.0 silicon camera [13], two 10 nm full-width half-maximum band-pass filters centered at a wavelength of 950 nm (Thorlabs LTD. Ely, United Kingdom FB950-10) and a Sigma imaging lens (Sigma Imaging (UK) Ltd. 13 Little Mundells, Welwyn Garden City, Hertfordshire, United Kingdom Macro 180) with a 180 mm focal length. A FOV of approximately 12 × 6 mm with an image size of 1024 × 512 pixels framed the electrode weld pool with a sufficiently high magnification for detailed analysis of the weld pool and with sufficient FOV to capture the cooling trail of the deposited material. The camera FPA was sensitive to a longest (cut-off) wavelength of approximately 1 µm. The two 950 nm filters were mounted sequentially along the optical axis and served to filter visible wavelengths, whilst providing blocking of the intense welding incandescence. A piece of shade 9 opaque acrylic welding shield was mounted between the camera and the work piece for mechanical protection of the lens in the case of projectiles (welding spatter) produced by the experiment. We determined, during calibration, that the filter had no effect upon optical transmission between 0.85 and 1.1 µm.

Two video modes were used, conditional upon the requirements of each experiment. These were 80 and 160 fps. The raw output from the camera was a series of 16-bit greyscale images within which the digital level (DL) of each pixel represented the measured NIR radiance from the scene. There is a concomitant increase in thermally generated NIR emission from any object with its thermodynamic temperature; this relationship can be modelled according to Planck’s law. Once this model is established for a sensor system, a calibration can be performed to fit the model to the properties of the as-built instrument [14]. Calibration, color mapping and other image processing steps were undertaken in MATLAB.

### 2.3. Calibration

Our camera system was calibrated with reference to a furnace designed for the purpose of calibrating radiation thermometers. The source of approximate blackbody radiation was a Landcal (Land Instruments, Dronfield, United Kingdom) R1500T which functioned as a cavity emitter with an emissivity of 0.99 at NIR wavelengths, according to the manufacturer data sheet [15]. The calibration was performed at a distance of 500 mm from the furnace, which was a comparable imaging distance to the distance from camera to blade surrogate in the experimental welding. There is not an agreed upon method by which thermal imaging cameras should be calibrated, and thus, our calibration was performed, where possible, by analogy to standard practices in radiation thermometry [16].

The furnace calibration assumed a black body emissivity (an emissivity of 1). As such, the camera system used in this work gave radiance temperature, rather than a truly absolute temperature. Accurate emissivity calculation of materials is difficult, especially when undergoing phase changes, and carries large uncertainly [17]. For this reason, it is considered good practice to work with radiance temperatures during analysis processes using IR thermal data and to apply emissivity to the final result [14]. Therefore, all temperature values presented in this work are radiance temperatures. A temperature error can be assigned to these temperatures according to equation 3.1 presented by Hobbs et al. [18]. The center wavelength of the system was calculated to be 952 nm based on the camera sensitivity and filter transmission. Using this value and a temperature of 1550 °C, the formula gives a temperature error of 0.45% per °C.

An emissivity value can be treated as a percentage and applied to the value calculated above to find an approximate error between the radiance temperatures presented here and the absolute temperature. This is because emissivity is expressed as a value between 0 and 1. A value for the emissivity of molten stainless steel at 0.27 can be found in the literature [19]. This is multiplied by 100 (to use the value as a percentage), then divided by the error per °C to give the approximate temperature error. Using the example temperature above, this value is 59 °C. This process can be repeated with any temperature presented in this work where the read feels necessary

A series of images was taken during calibration at temperatures from 700 to 1500 °C. Temperatures below 700 °C are difficult to measure due to the lack of signal because of the silicon sensor’s lack of sensitivity to wavelengths beyond 1100 nm. The bit depth of the images enabled measurements from 700 to 2400 °C. The sensor was assumed to be linear, and, therefore, the Planck model could be used to extrapolate to the maximum temperature of the measurement system. The temperature of the calibration furnace was verified at calibration temperatures using a transfer standard radiation thermometer (Land Cyclops C100L [20]) that had a valid calibration certificate and could be traced to the SI [21]. The mean average image of each calibration temperature was assembled to reduce noise in the calibration images. The Sakuma–Hattori method [22] was then applied to the central pixel DL within the aperture to create a signal-to-temperature calibration function for the camera system. The resulting temperature calibration curve can be seen in Figure 2.

### 2.4. Analysis

Our approach to thermal imaging analysis of the molten weld pool was by means of detecting the plateau region, formed by freezing of the weld pool, along a line within the temperature image. The plateau is due to the latent heat of freezing of the liquid, where energy is released due to crystal formation and briefly alters the rate of cooling of the liquid as a function of time [23]. The freezing plateau temperature of pure metals are well known, and these are used to define the temperature scale ITS-90 and provide a procedure for realizing the practical implementation of the Kelvin [16]. However, identifying the plateau can be very difficult, particularly for impure liquids. Therefore, another method was required to reliably detect the solidification point along the full height of the weld.

A multiple thresholding method was used to take advantage of having a relatively large area of temperature data within the weld. This method first cropped the image to remove the electrode from the FOV whilst also removing the cold surroundings. It then thresholded the image in a binning process, storing only those temperatures above the threshold temperature. This was repeated for each threshold temperature every 5 °C across the temperature range of the camera, creating a series of binned images. The area in pixels of each binned image was then plotted against its threshold temperature, and the gradient of the resulting curve was calculated. Peaks on the gradient curve were then taken as potential places for the solidification point (Figure 3c, potential peaks highlighted with arrows). A peak was then selected as the most likely location for the solidification point based on its proximity to the last point and the lowest absolute value of the peak. This function was weighted to favor peaks closer to the last because the solidification point did not move by large amounts between frames due to the high frame rate of the camera.

The solidification point along the weld track was determined using the red peak that corresponded with the threshold image (Figure 3c). The left-hand edge of the threshold area was used to determine the starting point for solidification in the weld track. However, these data were cleaned and smoothed using fill, close and erode digital image processing morphological operations from MATLAB’s Image Processing Toolbox. The fill operation filled in any gaps in the threshold image, and the close operation started to smooth the edges of the thresholded area. The close operation was a combination of an erosion operation, which set the current pixel value to the minimum values seen within the tool area (here, a rectangle of 5 × 2 pixels) followed by a dilation operation, which is the inverse, and set a pixel to the maximum value seen within the tool. The mask was then eroded again. The overall effect of this was to remove small light or dark spots from the mask (Figure 3d).

The leftmost pixel was then determined to be the solidification point of each row. The coordinates of which were recorded, and finally a moving average was run over the list of y coordinates to remove high-frequency movements which were likely to be an artefact of the image-processing steps and noise in the original image. The final points can be seen plotted on the temperature image in Figure 4.

## 3. Results and Discussion

A calibrated thermal imaging video exhibiting the initial root pass of the welding process is presented in Appendix A of which Figure 5 is a still frame. Various features of the weld pool dynamics can be seen in the video. Notably, the initial area of welding that consumes the wire can be seen to go through a freezing transition. The thermally driven, quasi-cyclic flow of material can be seen to introduce anisotropies into the weld. Fluid dynamics on the molten metal are especially evident during the wire withdrawal. Further characteristics of the material, such as viscosity, could be potentially inferred from this motion afforded by the high-resolution of the imaging system.

The dynamic range of a greyscale display is typically 8-bit. Therefore, it was necessary to represent the levels of grey artificially in different colors in order to best visually represent the data for interpretation. The Cube Helix color map was chosen [24]. This increased the dynamic range of the thermal video to 11 bits, as viewed on our display.

A good approximation of the cooling rate can be made from the images due to the profile view of the welding process. The molten material’s position varied with time, and the temperature of the material varied with time as it cooled. This meant that a line plot of temperature (spatial temperature plot) across the weld from right (the molten weld pool) to left (the solidified material) indicated the freezing point of the metal, which could be identified by a decrease in the gradient of the spatial temperature plot. However, the freezing point became harder to distinguish the closer the line profile was taken to the substrate. We believe this was due to the greater heat sinking effect caused by the proximity to the substrate. The relatively high level of noise in the signal made the freezing plateau less well defined than we would expect during an ITS-90 calibration. An example profile demonstrating this can be seen in Figure 6b below.

Figure 6 above demonstrates the need for high-resolution thermal imaging of the welding pool. The thermal history varies along the vertical profile of the weld. The side-on view of the process coupled with both line profiles shows the interaction between the new material and existing material in terms of thermal dissipation.

The images showing the full weld track were analyzed using the analysis technique described in Section 2.4. A video of the frame-by-frame analysis can be seen in the Appendix A. This gave a history along the full weld track of the detected freezing point, with the work piece moving approximately 22.5 µm between each frame. A location of the freezing point in the image space could be calculated from this information, which is vital when analyzing the thermal history of a material during an additive manufacturing process. The mean X value of the curve in the image space was then used to plot the location of the freezing point as a function of time. These data were low-pass filtered to remove high-frequency spikes in the location caused by the algorithm detecting the edges of patches of solid material that we considered to be slag, floating on top of the molten pool.

Figure 7, above, plots the freezing point of the weld with time. The anomalous initial value in the data is an artefact of the low-pass filter used. Data peaks at 1 and 2 Hz are seen when analyzing the frequency content of the mean X location. This frequency correlates to 0.9–1.8 mm movement of the turbine blade surrogate. This scale of feature can be seen on the weld bead in multiple places across the blade surrogate in Figure 8.

Figure 8 shows that the majority of the weld bead, especially towards the top edge, was smooth; this is considered a good visual indicator for a good weld. However, ripples can be seen in places towards the lower edge and across the full bead, which is an indicator of a possible weld defect. The ripples corresponded to the size of the feature detected by the freezing point analysis. This shows potential defects being detected with this analysis method. There are multiple reasons why a ripple can form in a weld bead, most of which relate to the imbalance of pressure in the molten weld pool. There have been multiple explanations for this, ranging from the composition of the molten material at that specific point, power supply oscillations and variations, and variations in solidification growth rate due to thermal variation [25,26].

It is possible that the seam that separates the flat surface of the blade surrogate and the freshly added material could be made uniform by online feedback to the welding rod feed from the welding pool dimensions obtained through our thermal imaging.

## 4. Conclusions

We have demonstrated that high spatial and temporal resolution thermal imaging can provide valuable, novel insights into the real-time material properties of the TIG welding process. We can provide welding pool dimensions with high spatial accuracy using the calculated phase transition point throughout the welding process. This information could be fed back into the process to help prevent defect formation and improve the quality of welds. For example, real-time temperature information could allow process parameters such as weld rod feed rate or arc current to be varied. Our post-build analysis has also shown macroscopic defects for which we offer an explanation and potential for a solution through feedback controls. This work lays the foundations for further investigation and analysis of the weld pool dynamics using our imaging technique that provides unprecedented spatial and temporal resolution of molten metal flow dynamics on the millimeter scale. Future work could involve analysis of the fluid dynamics of the melt pool, given that the motion of the liquid metal is so apparent in the images.

## Figures and Tables

**Figure 1 sensors-20-06952-f001:**
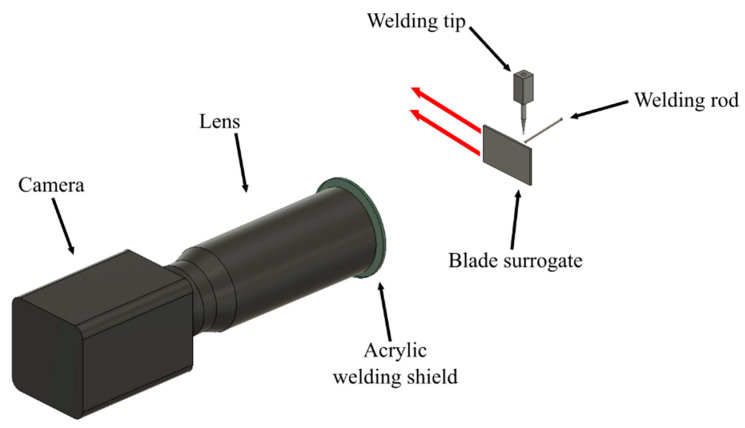
Representation of the experimental setup. The red arrows represent the direction of motion of the blade surrogate.

**Figure 2 sensors-20-06952-f002:**
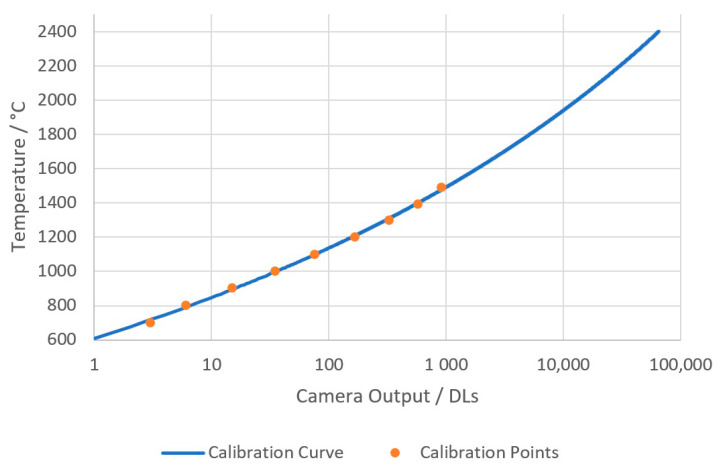
The temperature calibration curve with the calibration points overlaid.

**Figure 3 sensors-20-06952-f003:**
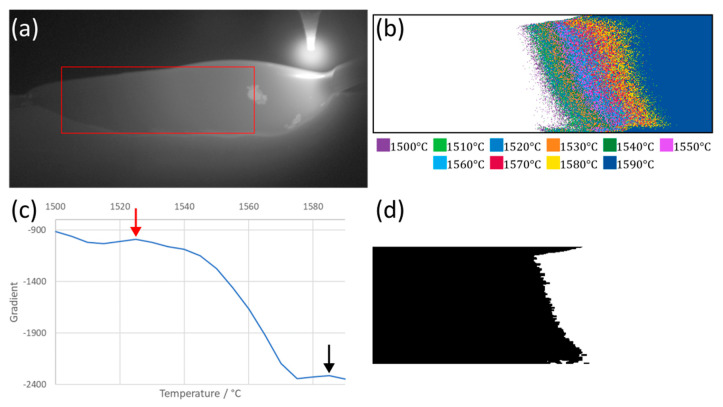
Progression of the analysis for a single still frame. A video of this analysis can be seen in Appendix A. (**a**) A still thermal image frame with the cropped region highlighted in red. (**b**) The multiple thresholded cropped regions. Each temperature range represented by a different color for visual clarity (performed every 5 °C, shown every 10 °C). (**c**) The gradient of the area in pixels of the binned images with peaks of the gradient highlighted with arrows. (**d**) The mask produced by taking the highlighted point in red from (**c**). The black region is solid material, and the white represents molten material.

**Figure 4 sensors-20-06952-f004:**
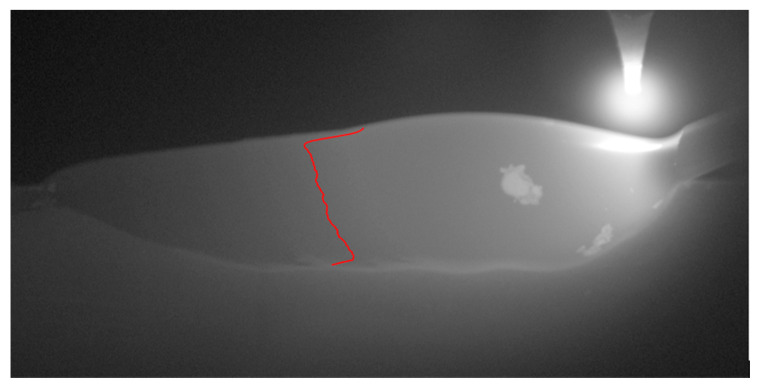
A still frame from the thermal imaging with the phase boundary overlaid in red.

**Figure 5 sensors-20-06952-f005:**
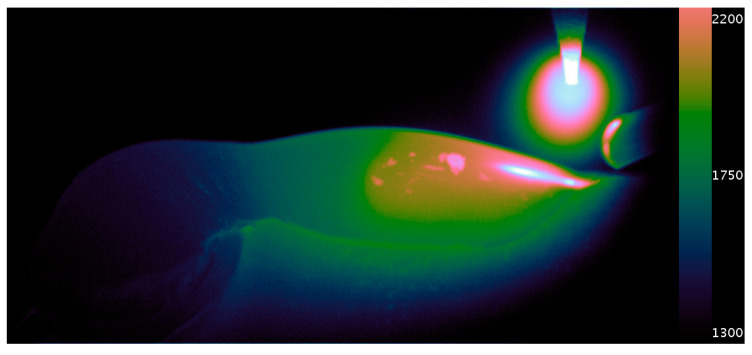
An example frame from the thermal video (scale in °C) with the Cube Helix color map applied. Slag floating on the molten weld can be seen in the center of the image, giving an indication of the weld pools dynamics. The heat transfer into the workpiece and the cooling regions of solidified weld can also be seen towards the bottom left of the image.

**Figure 6 sensors-20-06952-f006:**
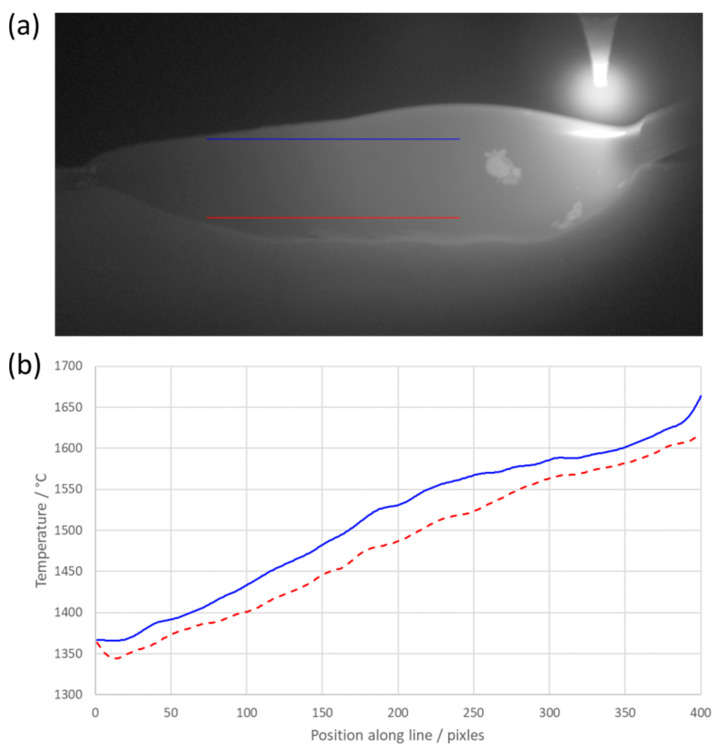
Temperature profile lines of the weld pool: (**a**) a still frame from the thermal imaging with both line profiles overlaid; (**b**) a graphical representation of the temperature profiles varying with position. The color of the lines corresponds to the color of the line profiles in a).

**Figure 7 sensors-20-06952-f007:**
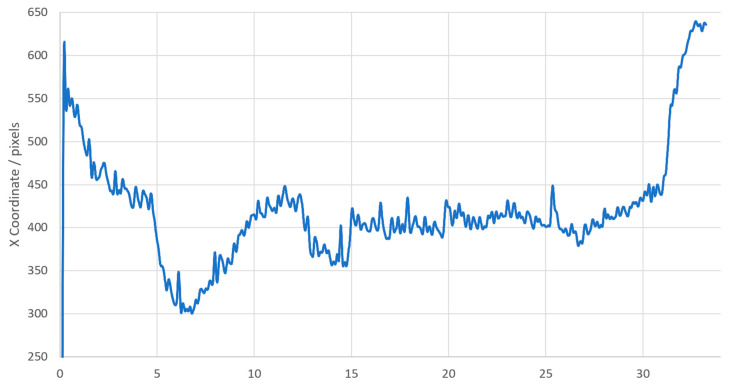
The x coordinate of the freezing point varying with time.

**Figure 8 sensors-20-06952-f008:**
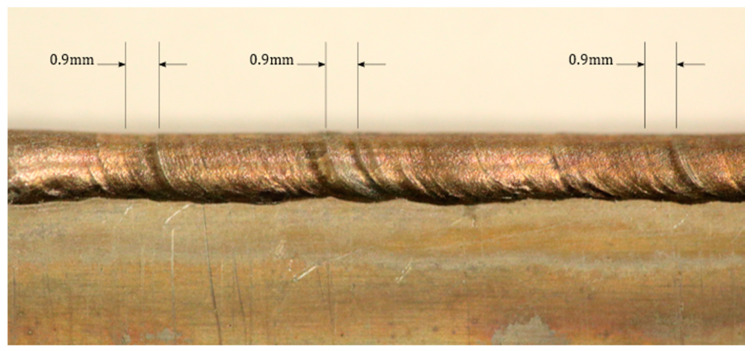
A high-resolution photograph of the blade surrogate showing the detected features from the thermal imaging.

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
