# Peer review of "High-Resolution Thermal Imaging and Analysis of TIG Weld Pool Phase Transitions"

_sensors, 2020, doi:10.3390/s20236952_

Round 1
Reviewer 1 Report
The work is interesting. However, the aoverall contribution and quantitative analysis need to be improved significantly as list below.
- More background work including weld material characterisation using non-destructive test and evaluation (NDT&E) should be reviewed;
- Section 3 should provide discussion of emissivity influence linking with section 2.3 /light spot, and quantitative analysis of weld material characterisation and comparison for sample in Figure 8. e.g.
Y Gao, GY Tian, Emissivity correction using spectrum correlation of infrared and visible images, Sensors and Actuators A: Physical 270, 8-17, 2018.
- Mind future work.
Author Response
Thank you for your feedback. Your point have been addressed and are detailed below:
1. More background work including weld material characterisation using non-destructive test and evaluation (NDT&E) should be reviewed;
Lines 56-65 discuss other NDT&E technique and the relative benefits of our technique.
2. Section 3 should provide discussion of emissivity influence linking with section 2.3 /light spot, and quantitative analysis of weld material characterisation and comparison for sample in Figure 8.
Lines 158-174 discuss emissivity correction and the benefits of radiance temperature over absolute temperature for analysis of this kind. It includes calculations of temperature error.
3. Mind future work.
Lines 321 and 322 set out potential future work.
Reviewer 2 Report
The present manuscript by Nicholas Boone, Matthew Davies, Jon Raffe Willmott, Hector Marin-Reyes and Richard French, entitled "High resolution thermal imaging and analysis of TIG weld pool phase transitions", describes a new approach to the real-time quality monitoring of a welding process, based on measurement of the weld temperature by registering its radiation intensity in the nearinfrared range. By the reviewer's estimation, the topic of the described research is highly relevant and potentially has a high practical applicability. The manuscript is generally written in a logical and comprehensive manner. The chapter 'Introduction' provides a convincing explanation of the necessity for this study and presents the experimental approach clearly and understandably. The chapter 'Materials and Methods' contains a detailed description of the experimental procedure, while the chapter 'Results and discussion' gives a holistic overview of the achieved results. The chapter 'Conclusions' summarizes the findings and the future perspectives in a concise, but comphehensive way. All the references are relevant, despite the unusually high number of cited webpages. English is perfect, and the manuscript itself is highly readable.
All-in-all, the reviewer does recommend this manuscript for publishing, after few minor corrections are possibly introducted.
1. It could be specified, whether this method is feasible for temperature measurements below 800 deg[C].
2. It would be nice to learn, how much time it takes for MATLAB to analyse the picture using a definite hardware setup.
3. Lines 49, 52: the abbreviations 'AM' and 'CNC' could be explained here.
4. Line 53: what was meant under "mechanical imperfections"? Contamination of what (filler metal, welded metal, etc.) was meant here?
5. Lines 95, 114, 258, 259, 260: the numerical values should be splitted from the measuring units.
6. Line 171: was the melting point meant here under "threshold temperature"?
7. Line 322: the journal title should be written in the title case.
8. Line 331: the journal title should be written in full.
9. Line 335: the journal title, volume, issue and page no.-s should be added here.
Author Response
1. It could be specified, whether this method is feasible for temperature measurements below 800 deg[C].
Lines 176 and 177 discuss temperature measurement below 700 degC.
2. It would be nice to learn, how much time it takes for MATLAB to analyse the picture using a definite hardware setup.
We did not address this point because it did not fit within the scope of the analysis description.
3. Lines 49, 52: the abbreviations 'AM' and 'CNC' could be explained here.
Line 35 and 52 address these abbreviations respectively.
4. Line 53: what was meant under "mechanical imperfections"? Contamination of what (filler metal, welded metal, etc.) was meant here?
Lines 53 and 54 edited to better explain the meaning of the mechanical imperfections and contamination.
5. Lines 95, 114, 258, 259, 260: the numerical values should be splitted from the measuring units.
All units spaced from numerical values.
6. Line 171: was the melting point meant here under "threshold temperature"?
Lines 203 and 204 edited to be clearer on this point. We do believe that threshold temperature is adequately explained in section 2.4
7. Line 322: the journal title should be written in the title case.
8. Line 331: the journal title should be written in full.
9. Line 335: the journal title, volume, issue and page no.-s should be added here.
All references reformatted to comply with journal requirements.
Reviewer 3 Report
The manuscript entitled ‘High resolution thermal imaging and analysis of TIG weld pool phase transitions’ falls within the scope of the journal Sensors. The paper contains interesting experimental results as well as measurement procedures. It is of sufficient scientific interest and has originality in its technical content to merit publication. The authors have cited the relevant literature. Methods, interpretations of results are correct. The authors presented extensive material supporting the conducted research. The issues were well presented. In terms of content, the analysis does not raise any objections. The arrangement of work maintains substantive continuity and constitutes a logical whole. The proposed high spatial and temporal resolution thermal imaging may be useful in the analysis of weld formation. However, the manuscript needs some minor corrections.
Comments and remarks are presented below.
- Figure 1 should be inserted into the text close to their first citation (on page 3).
- Figures 1, 2 and 4 should be reduced.
- Figure 3 caption should be properly formatted.
- The Figure 6 caption should be redrafted. For example: Temperature profile lines of weld pool (or molten zone): (a) a still frame from the thermal imaging with both line profiles overlaid; (b) a graphical representation according thermal image.
- Conclusions should be bulleted.
- The conclusion 'This information could be fed back into the process to help prevent defect formation and improve the quality of welds.' should be explained. Which information and how (to prevent what defects) can be used? Apart from macroscopic defects, which are identified by other NDT methods, the reasons for their formation are well known
- Authors need to correct References. Please use abbreviated journal name.
- Authors should carefully check the manuscript as required by the Instructions for Authors.
Author Response
1. Figure 1 should be inserted into the text close to their first citation (on page 3).
Figure 1 moved as requested.
2. Figures 1, 2 and 4 should be reduced.
These figures have been reduced.
3. Figure 3 caption should be properly formatted.
Figure 3's caption has been reformatted to comply with the journal specifications.
4. The Figure 6 caption should be redrafted.
Figure 6 caption has been redrafted as requested.
5. Conclusions should be bulleted.
The conclusion is not required to be bulleted and so is unchanged because this was not requested by the other reviewers, and we feel that continuous prose suit the style of our conclusion better than discrete bullet points.
6. The conclusion 'This information could be fed back into the process to help prevent defect formation and improve the quality of welds.' should be explained. Which information and how (to prevent what defects) can be used? Apart from macroscopic defects, which are identified by other NDT methods, the reasons for their formation are well known
Lines 56-65 and 315-316 detail the advantages of the technique and provide examples of how process parameters could be varied to prevent defect formation based on the output of our analysis.
7. Authors need to correct References. Please use abbreviated journal name.
All the references have been corrected to comply with journal requirements.
8. Authors should carefully check the manuscript as required by the Instructions for Authors.
We consulted the guidance to ensure that the formatting was corrected as required.
Round 2
Reviewer 1 Report
The improvement is reasonably good. More critical discussion is expected. For example, to address emissivity issue, different approaches including normalisation and correction should be discussed. e.g. Y Gao, etc, Emissivity correction using spectrum correlation of infrared and visible images, Sensors and Actuators A: Physical 270, 8-17, 2018.
Mind reference usage e.g. Ref. 17 is incomplete. Refinement or replacement of ref. 20-24 as they are out of date and unable to be accessed.
Author Response
Line 365 corrected refence issue and page numbers.
Line 380 Replaced refence for colour map.
The other references identified are the only sources appropriate for the material we cover in the manuscript.
We leave the emissivity section unchanged. We believe we have satisfactorily calculated the error due to emissivity in our work and appropriately justified why emissivity correction is not necessary for the technique we have used. We do not discuss methods of emissivity correction in the paper that you suggest we reference for this reason.